# Development of a checklist of standard items for processing individual participant data from randomised trials for meta-analyses: Protocol for a modified e-Delphi study

**Kylie E. Hunter**[1]*, **Angela C. Webster**[1], **Mike Clarke**[2], **Matthew J. Page**[3], **Sol Libesman**[1], **Peter J. Godolphin**[4], **Mason Aberoumand**[1], **Larysa H. M. Rydzewska**[4], **Rui Wang**[5], **Aidan C. Tan**[1], **Wentao Li**[5], **Ben W. Mol**[5], **Melina Willson**[1], **Vicki Brown**[6], **Talia Palacios**[1], **Anna Lene Seidler**[1]

**1** NHMRC Clinical Trials Centre, The University of Sydney, Camperdown, New South Wales, Australia, **2** School of Medicine, Dentistry and Biomedical Sciences, Centre for Public Health, Queen's University Belfast, Belfast, Northern Ireland, United Kingdom, **3** Methods in Evidence Synthesis Unit, School of Public Health and Preventive Medicine, Monash University, Melbourne, Victoria, Australia, **4** MRC Clinical Trials Unit at University College London, London, United Kingdom, **5** Department of Obstetrics and Gynaecology, Monash Health, Monash University, Melbourne, Victoria, Australia, **6** Deakin Health Economics, Institute for Health Transformation, School of Health and Social Development, Faculty of Health, Deakin University, Burwood, Victoria, Australia

* kylie.hunter@sydney.edu.au

**Data Availability Statement:** No datasets were generated or analysed during the current study. All

## Abstract

Individual participant data meta-analyses enable detailed checking of data quality and more complex analyses than standard study-level synthesis of summary data based on publications. However, there is limited existing guidance on the specific systematic checks that should be undertaken to confirm and enhance data quality for individual participant data meta-analyses and how to conduct these checks. We aim to address this gap by developing a checklist of items for data quality checking and cleaning to be applied to individual participant data meta-analyses of randomised trials. This study will comprise three phases: 1) a scoping review to identify potential checklist items; 2) two e-Delphi survey rounds among an invited panel of experts followed by a consensus meeting; and 3) pilot testing and refinement of the checklist, including development of an accompanying R-markdown program to facilitate its uptake.

## Introduction

### Background and rationale

Individual participant data meta-analyses (IPD-MA) involve the central collection of data for each individual participant in each study included in a systematic review. They offer many advantages over aggregate data meta-analyses, and can provide key evidence to inform guidelines, policy and practice [1]. However, the validity of any meta-analysis depends on the quality

relevant data from this study will be made available upon study completion.

**Funding:** The author(s) received no specific funding for this work.

**Competing interests:** I have read the journal's policy and the authors of this manuscript have the following competing interests: KEH receives research funding support via two scholarships administered by the University of Sydney (Postgraduate Research Supplementary Scholarship in Methods Development (SC3504), and Research Training Program Stipend (SC3227)). ALS is co-convenor and KEH & ACW are associate convenors of the Cochrane Prospective Meta-analysis Methods Group. MJP is recipient of the Australian Research Council Discovery Early Career Researcher Award (DE200101618), co-convenor of the Cochrane Bias Methods Group, and President of the Association for Interdisciplinary Meta-research and Open Science. RW is recipient of a National Health and Medical Research Council Investigator Grant. VB is supported by an Alfred Deakin Postdoctoral Research Fellowship. MC is co-convenor (unpaid) of the Cochrane Individual Participant Data Meta-analysis Methods Group; LHMR is coordinator of this group; KEH, PJG, BWM, MC and ALS are members. LHMR is supported by the UK Medical Research Council (https://protect-au.mimecast.com/s/iksjCK1DvKTqkx5mBl3AxP2?domain=mrc.ukri.org) Grant number: MC_UU_00004/06. ALS is recipient of a National Health and Medical Research Council Investigator Grant. BWM is recipient of a National Health and Medical Research Council Investigator grant (GNT1176437), reports consultancy for ObsEva at an hourly rate, reports consultancy for Merck Merck KGaA at an hourly rate and received travel support from Merck Merck KGaA. This does not alter our adherence to PLOS ONE policies on sharing data and materials.

of the included studies and their data [2, 3]. One advantage of IPD-MA is that access to raw data allows more detailed data quality checks than are possible when data are synthesised at study level, in aggregate. Addressing any issues identified by these checks enables higher quality data to be included in the meta-analysis which leads to more robust and reliable results. Hence, data quality checks are considered an integral component in the conduct of IPD-MA [4].

Despite this, a recent systematic review evaluating the methodological conduct of all IPD-MA of randomised trials published in English up to September 2019 (n = 323) reported that only 56% conducted checks for invalid, inconsistent, out of range or missing data [5]. This may be due to inadequate resourcing and under-appreciation of the time-intensive nature of such checks [6], which can require at least three to four days per trial and up to several weeks for large and complex datasets [7]. Furthermore, while there is an abundance of literature providing advice on the different types of data quality checks that could/should be conducted, it is not easily actionable. Guidance on which checks should be systematically performed and *how* these checks should be conducted is lacking. For instance, in the IPD-MA Handbook for Healthcare Research [4], the authors advise that validity, range and consistency of variables should be checked, and recommend using a standardised approach across data checkers, such as following a common checklist and/or statistical code developed for this purpose. However, the specific items on this checklist and how to action them are not elucidated. The PRIME-IPD tool [6, 8, 9] provides a useful framework for preparing IPD for meta-analysis encompassing five steps (Processing, Replication, Imputation, Merging, and Evaluation), but is not prescriptive about how each step should be performed [6]. Authors of PRIME-IPD have acknowledged that the framework needs to be refined and suggest a consensus approach such as a Delphi survey method would be appropriate for this purpose [6, 9].

## Objectives

We aim to address this gap by developing an internationally recognised checklist to guide routine data quality checking and cleaning for IPD-MA of randomised trials: the CHecklist for Individual Participant data Processing of Randomised Trials (CHIPPR). While the focus is on randomised trials, many of the checks will also apply to data from other types of studies being brought together for secondary use, such as observational studies. Fig 1 depicts how CHIPPR fits within the overall conduct of an IPD-MA, and distinguishes between three separate but related types of checks that should be undertaken. One type of checks focuses on integrity and identification of potential research misconduct (fabrication, falsification, or plagiarism) [10], which is not within scope of this study and for which a checklist (informed by a scoping review) [11] is already in development. Another type is risk of bias checks, which can be informed by direct checking of IPD, e.g. to assess whether randomisation is robust. The third type is data quality checks and this forms the focus of this paper.

We define data quality as the extent to which dimensions of data meet requirements, where a dimension is a measurable feature of a data concept such as accuracy, completeness, consistency and validity (Table 1) [12, 13]. The three types of data checks overlap in Fig 1 to demonstrate that some checks are common, e.g. identification of implausible dates is a common component across data quality and integrity checks; and checking for imbalance in baseline characteristics is common for risk of bias assessments, integrity checks and data quality. Each checklist item will include methods of assessment and decision criteria to guide management of any issues identified. To facilitate uptake of CHIPPR, we will develop an R-markdown program [14–16] to automate some of the processes described in the checklist. This may be

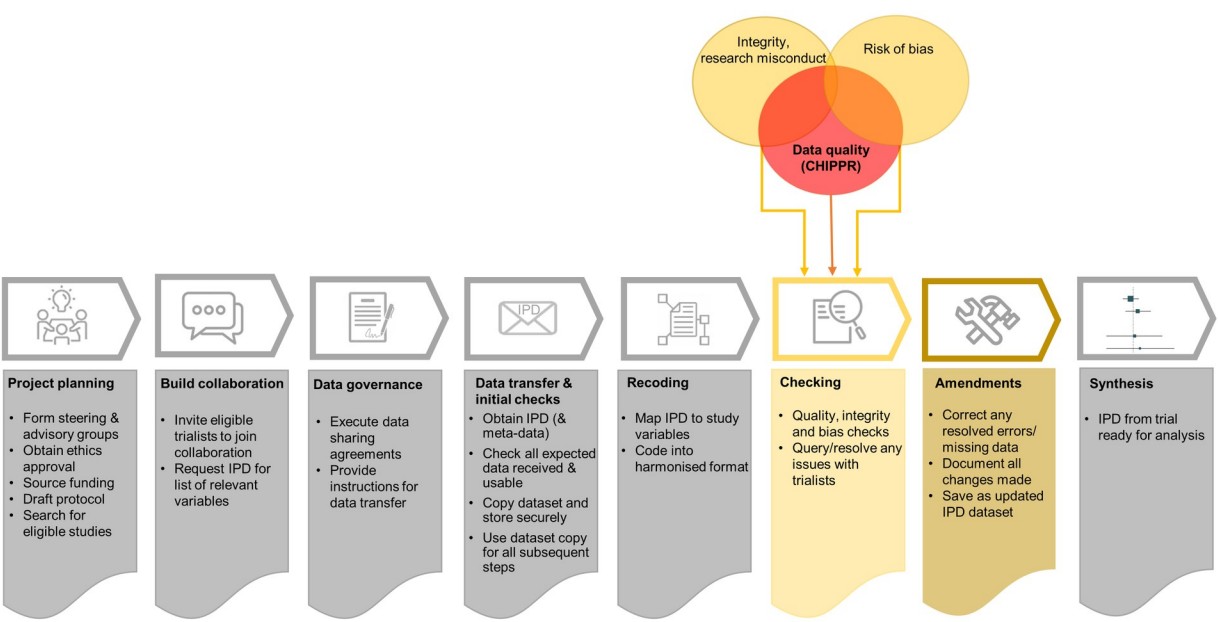

**Fig 1. Stages of an individual participant data meta-analysis, highlighting where CHIPPR fits in and interrelated types of data checks (adapted from Rydzewska & Tierney)** [4]**.**

adapted to other statistical software (e.g. STATA, SAS) in the future. The ultimate objective is to improve the quality and reliability of IPD-MA.

## Materials and methods

As shown in Fig 2, this study will comprise three phases: 1) a scoping review to identify potential checklist items; 2) two e-Delphi survey rounds among an invited panel of experts followed by a consensus meeting; and 3) pilot testing and refinement of the checklist, including development of an accompanying R-markdown [14–16] program to facilitate its uptake. The project will be coordinated and overseen by an international steering group of experts in IPD methodology, evidence synthesis, statistics, clinical trial design and data management.

### Phase 1: Scoping review to identify potential checklist items

We will undertake a scoping review in accordance with Joanna Briggs Institute (JBI) guidelines [17], to determine which data quality checks researchers perform when processing datasets for inclusion in an IPD-MA of randomised trials. A full protocol for this review, covering each relevant item of the Preferred Reporting Items for Systematic Reviews and Meta-Analyses Extension for Scoping Reviews (PRISMA-ScR) [18], has been preregistered on the Open Science Framework (OSF, https://osf.io/g2unf/).

We will include systematic reviews with IPD-MA of randomised trials on intervention effects published in English. These have previously been identified up to September 2019 in a systematic review by Wang et al. [5], and we will update their search to include all others published up to July 2022. Records will be independently screened by two reviewers, with any discrepancies resolved by discussion or, if required, adjudication by a third reviewer. For each eligible record, we will obtain relevant supplementary materials that are attached or referred to in the publication, e.g. statistical analysis plan, data management plan, PROSPERO registration record. We will extract information on characteristics of the included IPD-MA (e.g. year

**Table 1. Dimensions of data quality.**

| Data dimension | Definition | Examples of non-compliance |
|---|---|---|
| Completeness | The degree to which i) all required variables, ii) all required records, and iii) all required data values are present in the dataset | • Birthweight variable is missing from dataset, but is reported in publication |
| | | • Not all randomised participants are present in the dataset |
| | | • Birthweight is not provided for every record/ participant |
| Reasonability | The degree to which a data pattern meets expectations. Includes time series trends, expectations of randomness and digit preference | • Allocation patterns not random |
| | | • Despite equal chance a digit may take any value, it is never 5 or mostly 8 |
| | | • Participant height at time 2 is less than height at time 1 |
| Compliance | The degree to which values and variables are in accordance with IPD study codebook either in their original form or after transformation. Includes variable definition, format specification, categories and outcome scales | • Nutritional intake should be in kilojoules but is provided in calories |
| | | • Post-partum haemorrhage defined as blood loss >600ml rather than >500ml |
| | | • Date recorded as mm/dd/yy instead of dd/mm/ yyyy |
| | | • Sex should be coded male = 1, female = 2, but is female = 1, male = 2 |
| Consistency | The degree to which i) data values match corresponding publications or reports (external consistency), or ii) data values of two or more variables within a participant are logical or comply with a rule or equation (internal consistency) | • Publication reports 52/168 children had obesity; data shows 49/170 |
| | | • Protocol states eligible age is ≥18 years, but participant is 15 years old |
| | | • Participant body mass index does not equal their weight/height$^2$ |
| | Do not confuse consistency with accuracy or correctness | • Participant age at baseline was 50, but age at follow-up was 45 |
| | | • Hours slept at night plus hours slept during the day is > 24 hours |
| | | • Date of follow-up assessment occurs before date of enrolment |
| Validity | The degree to which dates or data values are within a pre-specified or valid range. A data value can be valid but not accurate | • An enrolment date that occurs outside of study start and end dates |
| | | • Score on a scale of 1–10 is 12 |
| | | • Gestational age at birth is 54 weeks (outlier), but valid range is 20–45 weeks |
| Plausibility | The degree to which data values match knowledge of the real world (this can be seen as a type of consistency). Values may be possible but not plausible | • 90% of participants died despite scoring well on measures of health |
| | | • Participant self-reported intense physical activity as 20hrs/day |
| | | • 80% of routine visits occurred on a public holiday |
| Uniqueness | The degree to which records occur only once in a dataset | • Duplicate participant identifiers |
| | | • Identical values for all variables except participant identifier for ≥2 records |

Adapted from Black et al. [12, 13].

of publication, country, number of included studies) and descriptions of any procedures used to check, clean, transform or prepare individual participant data for analyses, including any software used. We will use a standardised data extraction form which will be piloted by five independent reviewers and revised accordingly prior to commencing full extraction. Where

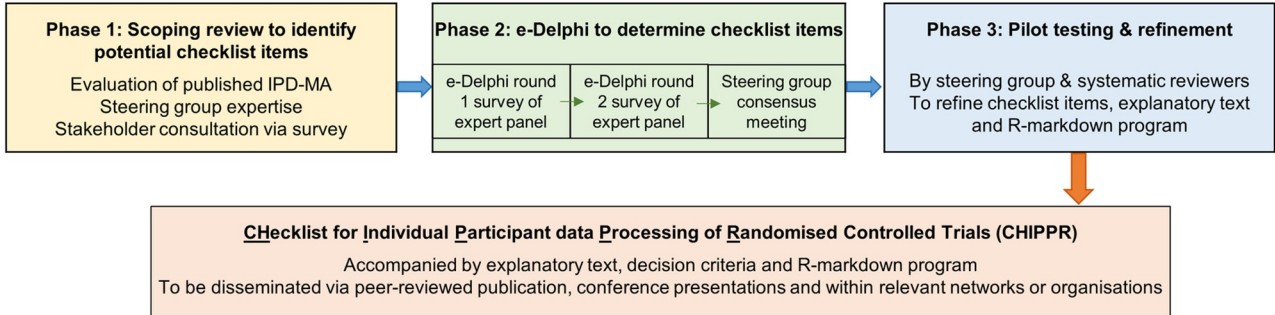

**Fig 2. Overview of project phases.**

available, the form will be pre-populated by data extracted by Wang et al. [5] to avoid duplication. Data will be extracted by one reviewer with a random subset of 10% of these checked by a second reviewer. Disagreements will be resolved through discussion, with a third reviewer involved if consensus is not reached. Extracted data will be presented in summary tables and figures.

Next, we will conduct a brief online survey of authors of these IPD-MA to obtain more detailed information about their data quality checking and cleaning processes. The survey will include the data already extracted for their IPD-MA (as above), and will ask authors to add any other data quality checks that are missing, with an emphasis on *how* these checks were performed. Reminder emails will be sent after two weeks to non-responders. All identified candidate items will be collated and grouped into common themes and domains (such as randomisation, inconsistencies and missing data) using inductive thematic analysis [19].

Lastly, the study team will draw on their expertise and experience as IPD-MA experts, data managers, statisticians, methodologists and clinicians to add new items, exclude items beyond the scope of the research question, and refine the wording and descriptions of each candidate item for inclusion in the first e-Delphi survey round. A patient representative with previous research experience will also be separately consulted to provide feedback on the list of items. Results of the scoping review will be reported in accordance with the PRISMA-ScR [20] and disseminated via peer-reviewed publication.

## Phase 2: e-Delphi survey and consensus meeting

We will conduct a two-round modified electronic Delphi (e-Delphi) survey [21] among researchers with expertise in data processing, to achieve consensus on the most important items for checking and cleaning an IPD dataset. The e-Delphi surveys will be managed using Qualtrics software [22] and pilot tested by steering group members before being launched to the expert panel (see below). Participation involves completion of survey round 1 and/or 2, and consent is implied by completion. Responses will be quasi-anonymous [23], meaning they will be anonymous to all but the lead researcher, who will be able to link responses to individuals via a unique identifier. This enables provision of feedback on previous individual responses in subsequent rounds, which is a core feature of the Delphi method [21]. Each survey round will be open for 4 weeks, and reminder emails will be sent after weeks 1 and 3 to enhance response rates. There will be a break of approximately 2–3 weeks between rounds.

**Participants for expert panel.** We will invite a representative sample of relevant international stakeholders (e.g. systematic reviewers, statisticians, clinical trialists, data managers and journal editors) with experience in data processing for IPD-MA to participate in the e-Delphi

survey rounds. Fluency in English is required since this is the language in which surveys will be conducted. Participants will be recruited using purposive and snowball sampling [21]. Specifically, by contacting corresponding authors of all IPD-MA identified in the scoping review (phase 1), via key contacts of steering group members, and via relevant professional networks and organisations, including the Cochrane Individual Participant Data Meta-analysis (https://methods.cochrane.org/ipdma/) and Prospective Meta-analysis (https://methods.cochrane.org/pma/) Methods Groups. There is no consensus on the ideal sample size for Delphi studies, though the decision should be based on complexity of the topic, the degree and diversity of expertise required and available resources [21]. To enhance generalisability of results and account for attrition, we aim to recruit a broad range of panellists across diverse geographical locations and from a variety of therapeutic areas. Invitations will be sent to publicly available email addresses and will include a Participant Information Sheet and a link to access round 1 of the e-Delphi survey.

**e-Delphi survey, round 1.**   Panel members will each be issued with an anonymous identifier and asked to rate each candidate item by importance using a 7-point Likert scale (1 = 'not at all important'; 2 = 'low importance'; 3 = 'slightly important'; 4 = 'neutral'; 5 = 'moderately important'; 6 = 'very important'; 7 = 'extremely important') [24]. This is based on the recommendation that four to seven response categories are optimal for Delphi studies in health research [25]. Each item will be accompanied by a brief description of conceptual logic and rationale for inclusion. If panellists feel they lack the requisite expertise or understanding to appropriately rate an item, they will be instructed to select the option 'Unable to rate' (rather than 4 = 'neutral', which would dilute results). These will be classed as NA and excluded from the analyses. Panellists will also have the opportunity to suggest any modifications to the items, propose additional items, and elaborate on their responses using free text.

Basic demographic information will also be collected to provide an overall profile of panellists and to give an indication of their representativeness, e.g. area of expertise, qualifications, experience, country of residence, gender. On survey close, we will do descriptive analyses of participant characteristics and item rating scores (frequencies, proportions, median, interquartile range) to determine whether consensus has been reached. While there are no recognised guidelines on what constitutes an appropriate level of consensus, we will use the following *a priori* definition, based on review of the literature and consultation among steering group members [21, 25–27]:

- Consensus to include an item: ≥75% of the panel rate the item as 'very important' or 'extremely important'.

- Consensus to exclude an item: ≥75% of the panel rate the item as 'not at all important' or 'low importance'.

- No consensus: all other combinations.

Qualitative data in the form of free text responses will be reviewed and narratively summarised. Any new suggested items or modifications will be discussed among the steering group to determine whether they are unique and relevant for inclusion in round 2.

**e-Delphi survey, round 2.**   Panellists will be invited to participate in the e-Delphi round 2 regardless of whether they completed round 1, to reduce the chance of false consensus, to mitigate attrition, and so that opinions of the original panel are more accurately represented [28]. Since new panellists may join at round 2, all previous items from round 1 will be included in the survey plus any new suggested items. For each item, panellists will be provided with their individual response (where applicable) and the median and interquartile range score for the whole panel from round 1, and given the opportunity to revise and re-rate these items in

round 2 according to the same Likert scale [21]. Results will be analysed to assess consensus using the same criteria as for round 1. Attrition bias arising from participant withdrawal between rounds will be explored by comparing response distributions of withdrawn to completing participants [29]. We will also conduct an analysis of whether the chosen items would change if we limited the data to respondents to round 1 only, round 2 only or both round 1 and 2.

**Consensus meeting.** On completion of the two e-Delphi survey rounds, a consensus meeting will be held virtually among steering group members. This will use a nominal group technique to decide on a final set of checklist items and accompanying explanatory text. The meeting will begin with a brief overview of the study background and aims, followed by a summary of results of the e-Delphi survey rounds. The main focus will then be on discussion of items for which consensus could not be reached, and attendees will have the opportunity to anonymously vote for inclusion or exclusion of each of these in the final checklist. Items for which ≥75% of attendees vote to include will be added to the final checklist. The meeting will be recorded to aid reporting.

## Phase 3: Pilot testing

Before pilot testing of the checklist, we will develop an R-markdown template to aid implementation of the checklist. Next, members of the steering group will pilot test the checklist and R-markdown by applying them to IPD-MA they are conducting. We will also invite a convenience sample of systematic reviewers who have experience in conducting IPD-MA to conduct pilot testing. The aim of this phase is to refine the items and explanatory text and to streamline and improve the R-markdown based on participants' feedback using an iterative process until agreement on the final R-markdown is reached among the steering group.

**Data management.** All data will be stored in a secure password-protected folder at the NHMRC Clinical Trials Centre, University of Sydney, which is only accessible by authorised research personnel. Data management will be in accordance with the *University of Sydney Data Management Policy 2014*.

**Dissemination.** The final checklist and accompanying R-markdown program will be widely disseminated via peer-reviewed, open-access publication, presentations and via relevant social media (e.g. Twitter). We will also use our connections within networks such as Cochrane to encourage rapid integration in international best practice guidelines for systematic reviews and use of the checklist by world-leading organisations in this field.

**Ethics.** Ethics approval is not required for the phase 1 scoping review, as confirmed by the Chair of The University of Sydney Human Research Ethics Committee. Once the e-Delphi survey has been designed (phase 2) we will seek appropriate ethics approval. This study has been pre-registered on the Open Science Framework.

## Discussion

The importance of performing data quality checks and cleaning for IPD-MA is widely recognised, but there is limited existing guidance on what this should entail. We will address this gap by developing a checklist of standard items that can be used by researchers to help ensure the highest quality data are included in their review. This will improve the quality of their review, reduce the risk that the results of meta-analyses will be biased and improve the evidence base for guidelines and clinical practice, ultimately improving health outcomes for the public.

Strengths of this study include use of scoping review methodology to develop initial survey items and use of e-Delphi survey methodology, which enables quasi-anonymity of participant

responses, global accessibility to facilitate greater sample size and representativeness, stream-lined data collection, reduced administrative costs and open sharing of opinions without undue influence from dominant individuals [30, 31]. Provision of controlled feedback between the two survey rounds also allows participants to re-consider their initial ratings, taking into account the views of other stakeholders [21]. Potential limitations include the possibility of a low number of participants and participant attrition. We will mitigate these risks by recruiting as many eligible participants as possible, clearly outlining the requirements of participation from the outset, sending reminder emails and scheduling only 2–3 weeks between rounds to maintain participant engagement [21, 25, 27].

## Patient and public involvement

As part of the scoping review, a patient representative with previous research experience will be consulted to provide feedback on the list of items.

## Supporting information

**S1 Checklist. PRISMA-P checklist (modified for scoping review).**
(DOC)

## Acknowledgments

We are grateful to Lesley Stewart, Jayne Tierney (co-convenors of the Cochrane IPD Meta-analysis Methods Group) and Lisa Askie (co-convenor of the Cochrane Prospective Meta-analysis Methods Group) for their input at the conceptualization phase of this study.

## Author Contributions

**Conceptualization:** Kylie E. Hunter, Anna Lene Seidler.

**Investigation:** Kylie E. Hunter.

**Methodology:** Kylie E. Hunter, Angela C. Webster, Mike Clarke, Matthew J. Page, Sol Libesman, Peter J. Godolphin, Mason Aberoumand, Larysa H. M. Rydzewska, Rui Wang, Aidan C. Tan, Wentao Li, Ben W. Mol, Melina Willson, Vicki Brown, Talia Palacios, Anna Lene Seidler.

**Project administration:** Kylie E. Hunter, Angela C. Webster, Mike Clarke, Matthew J. Page, Sol Libesman, Peter J. Godolphin, Mason Aberoumand, Larysa H. M. Rydzewska, Rui Wang, Aidan C. Tan, Wentao Li, Ben W. Mol, Melina Willson, Vicki Brown, Talia Palacios, Anna Lene Seidler.

**Supervision:** Angela C. Webster, Anna Lene Seidler.

**Writing – original draft:** Kylie E. Hunter.

**Writing – review & editing:** Kylie E. Hunter, Angela C. Webster, Mike Clarke, Matthew J. Page, Sol Libesman, Peter J. Godolphin, Mason Aberoumand, Larysa H. M. Rydzewska, Rui Wang, Aidan C. Tan, Wentao Li, Ben W. Mol, Melina Willson, Vicki Brown, Talia Palacios, Anna Lene Seidler.

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
