## [Decision Letter · Decision Letter 0]

31 Aug 2022

PONE-D-22-18844

Development of a checklist of standard items for processing individual participant data from randomised trials for meta-analyses: protocol for a modified e-Delphi study

PLOS ONE

Dear Dr. Hunter,

Thank you for submitting your manuscript to PLOS ONE. After careful consideration, we feel that it has merit but does not fully meet PLOS ONE’s publication criteria as it currently stands. Therefore, we invite you to submit a revised version of the manuscript that addresses the points detailed below that were raised during the review process.

We look forward to receiving your revised manuscript.

Kind regards,

Matthew Carroll, PhD., MEdL., MPod., BHSc

Academic Editor

PLOS ONE

Journal Requirements:

    "I have read the journal's policy and the authors of this manuscript have the following competing interests: KEH receives research funding support via two scholarships administered by the University of Sydney (Postgraduate Research Supplementary Scholarship in Methods Development (SC3504), and Research Training Program Stipend (SC3227)). ALS is co-convenor and KEH & ACW are associate convenors of the Cochrane Prospective Meta-analysis Methods Group. MJP is recipient of the Australian Research Council Discovery Early Career Researcher Award (DE200101618), co-convenor of the Cochrane Bias Methods Group, and President of the Association for Interdisciplinary Meta-research and Open Science. RW is recipient of a National Health and Medical Research Council Investigator Grant. VB is supported by an Alfred Deakin Postdoctoral Research Fellowship. MC is co-convenor (unpaid) of the Cochrane Individual Participant Data Meta-analysis Methods Group; LHMR is coordinator of this group; KEH, PJG, BWM, MC and ALS are members. LHMR is supported by the UK Medical Research Council (https://mrc.ukri.org/) Grant number: MC_UU_00004/06. ALS is recipient of a National Health and Medical Research Council Investigator Grant. BWM is recipient of a National Health and Medical Research Council Investigator grant (GNT1176437), reports consultancy for ObsEva at an hourly rate, reports consultancy for Merck Merck KGaA at an hourly rate and received travel support from Merck Merck KGaA."

Reviewers' comments:

Reviewer's Responses to Questions

**Comments to the Author**

1. Does the manuscript provide a valid rationale for the proposed study, with clearly identified and justified research questions?

Reviewer #1: Yes

2. Is the protocol technically sound and planned in a manner that will lead to a meaningful outcome and allow testing the stated hypotheses?

Reviewer #1: Yes

3. Is the methodology feasible and described in sufficient detail to allow the work to be replicable?

Reviewer #1: Yes

4. Have the authors described where all data underlying the findings will be made available when the study is complete?

Reviewer #1: Yes

5. Is the manuscript presented in an intelligible fashion and written in standard English?

Reviewer #1: Yes

6. Review Comments to the Author

You may also provide optional suggestions and comments to authors that they might find helpful in planning their study.

Reviewer #1: I thank the authors for contributing to the field of research quality and integrity. The authors highlighted the importance of the PRIME-IPD framework but also its limitation and they propose to develop a Delphi-informed checklist and a companion R-markdown program (strength of the project) to guide routine data quality checking and cleaning for IPD-MA of randomised trials. Overall, the manuscript is well-writing and documented.

I provided some comments, mostly related to phase 1-scoping review, to help improve the manuscript.

-Major

Page 6. Please provide the direct link of OSF to the registered protocol instead of the general OSF website so that the reviewers and readers can access the protocol.

-Minor

Page 7. The following statement “We will use a standardised data extraction form and check any available protocol publications, results publications and/or supplementary materials (e.g., statistical analysis plans, data management plans, PROSPERO registration records) for each included IPD-MA) is a bit confusing by combining the use of a standardised data extraction form and the process of looking for available protocol publications, etc. in the same sentence. For example, I do not know if the search of the PROSPERO database is part of a structured grey literature search or not. I was not able to get the OSF-registered protocol.

Page 9. The authors provided example of basic demographic information to be collected for representativeness: if gender is included, I think it’s worth mentioning in the list of examples.

Page 12. Patient and public involvement (PPI): I understand the explanation the authors, but I think PPI is necessary and they should try find and include them either in the panelist group or in the steering group.

Figures: please improve their resolution

7. PLOS authors have the option to publish the peer review history of their article (what does this mean?). If published, this will include your full peer review and any attached files.

Reviewer #1: No

---

## [Author Response · Author response to Decision Letter 0]

6 Sep 2022

Point-by-point responses:

Journal Requirements

Response: Thank you for sharing these links. The manuscript has been checked to ensure it meets all of PLOS ONE’s style requirements. 

 "I have read the journal's policy and the authors of this manuscript have the following competing interests: KEH receives research funding support via two scholarships administered by the University of Sydney (Postgraduate Research Supplementary Scholarship in Methods Development (SC3504), and Research Training Program Stipend (SC3227)). ALS is co-convenor and KEH & ACW are associate convenors of the Cochrane Prospective Meta-analysis Methods Group. MJP is recipient of the Australian Research Council Discovery Early Career Researcher Award (DE200101618), co-convenor of the Cochrane Bias Methods Group, and President of the Association for Interdisciplinary Meta-research and Open Science. RW is recipient of a National Health and Medical Research Council Investigator Grant. VB is supported by an Alfred Deakin Postdoctoral Research Fellowship. MC is co-convenor (unpaid) of the Cochrane Individual Participant Data Meta-analysis Methods Group; LHMR is coordinator of this group; KEH, PJG, BWM, MC and ALS are members. LHMR is supported by the UK Medical Research Council (https://mrc.ukri.org/) Grant number: MC_UU_00004/06. ALS is recipient of a National Health and Medical Research Council Investigator Grant. BWM is recipient of a National Health and Medical Research Council Investigator grant (GNT1176437), reports consultancy for ObsEva at an hourly rate, reports consultancy for Merck Merck KGaA at an hourly rate and received travel support from Merck Merck KGaA."

Response: I confirm that this does not alter our adherence to all PLOS ONE policies, and have updated the Competing Interests section as required:

"I have read the journal's policy and the authors of this manuscript have the following competing interests: KEH receives research funding support via two scholarships administered by the University of Sydney (Postgraduate Research Supplementary Scholarship in Methods Development (SC3504), and Research Training Program Stipend (SC3227)). ALS is co-convenor and KEH & ACW are associate convenors of the Cochrane Prospective Meta-analysis Methods Group. MJP is recipient of the Australian Research Council Discovery Early Career Researcher Award (DE200101618), co-convenor of the Cochrane Bias Methods Group, and President of the Association for Interdisciplinary Meta-research and Open Science. RW is recipient of a National Health and Medical Research Council Investigator Grant. VB is supported by an Alfred Deakin Postdoctoral Research Fellowship. MC is co-convenor (unpaid) of the Cochrane Individual Participant Data Meta-analysis Methods Group; LHMR is coordinator of this group; KEH, PJG, BWM, MC and ALS are members. LHMR is supported by the UK Medical Research Council (https://mrc.ukri.org/) Grant number: MC_UU_00004/06. ALS is recipient of a National Health and Medical Research Council Investigator Grant. BWM is recipient of a National Health and Medical Research Council Investigator grant (GNT1176437), reports consultancy for ObsEva at an hourly rate, reports consultancy for Merck Merck KGaA at an hourly rate and received travel support from Merck Merck KGaA. This does not alter our adherence to PLOS ONE policies on sharing data and materials."

Response: As requested, the ethics statement has been moved to the Methods section of our manuscript.

Response: A caption for the supporting information file (PRISMA checklist) has been added to the end of our manuscript. It is not cited in-text.

Response: The reference list has been reviewed and, to our knowledge, it is complete and correct. No changes have been made and no retracted articles have been cited. 

Reviewer’s comments to the Author 

Reviewer #1: I thank the authors for contributing to the field of research quality and integrity. The authors highlighted the importance of the PRIME-IPD framework but also its limitation and they propose to develop a Delphi-informed checklist and a companion R-markdown program (strength of the project) to guide routine data quality checking and cleaning for IPD-MA of randomised trials. Overall, the manuscript is well-writing and documented.

Response: Many thanks for this positive feedback. 

I provided some comments, mostly related to phase 1-scoping review, to help improve the manuscript.

-Major

Page 6. Please provide the direct link of OSF to the registered protocol instead of the general OSF website so that the reviewers and readers can access the protocol.

Response: Thank you for this suggestion. The direct link to the publicly available protocol has been added to the manuscript as follows:

“A full protocol for this review, covering each relevant item of the Preferred Reporting Items for Systematic Reviews and Meta-Analyses Extension for Scoping Reviews (PRISMA-ScR) (18), has been preregistered on the Open Science Framework (OSF, https://osf.io/g2unf/).” (p.7)

-Minor

Page 7. The following statement “We will use a standardised data extraction form and check any available protocol publications, results publications and/or supplementary materials (e.g., statistical analysis plans, data management plans, PROSPERO registration records) for each included IPD-MA) is a bit confusing by combining the use of a standardised data extraction form and the process of looking for available protocol publications, etc. in the same sentence. For example, I do not know if the search of the PROSPERO database is part of a structured grey literature search or not. I was not able to get the OSF-registered protocol.

Response: Thank you for this comment. We have re-worded relevant sections of this paragraph for clarity:

“We will include systematic reviews with IPD-MA of randomised trials on intervention effects published in English. These have previously been identified up to September 2019 in a systematic review by Wang et al (5), and we will update their search to include all others published up to July 2022…

… For each eligible record, we will obtain relevant supplementary materials that are attached or referred to in the publication, e.g. statistical analysis plan, data management plan, PROSPERO registration record... 

…We will use a standardised data extraction form which will be piloted by five independent reviewers and revised accordingly prior to commencing full extraction.” (p.7-8)

Page 9. The authors provided example of basic demographic information to be collected for representativeness: if gender is included, I think it’s worth mentioning in the list of examples.

Response: We have added gender as a demographic variable for collection (p.10).

Page 12. Patient and public involvement (PPI): I understand the explanation the authors, but I think PPI is necessary and they should try find and include them either in the panelist group or in the steering group.

Response: Thank you for this suggestion. We have invited a patient representative to participate in this project. This is articulated in the revised manuscript as follows:

“A patient representative with previous research experience will also be separately consulted to provide feedback on the list of items.” (p.9)

“Patient and public involvement

As part of the scoping review, a patient representative with previous research experience will be consulted to provide feedback on the list of items.” (p.14)

Figures: please improve their resolution

Response: The resolution of both figures has been improved to 300 dpi.

---

## [Decision Letter · Decision Letter 1]

26 Sep 2022

Development of a checklist of standard items for processing individual participant data from randomised trials for meta-analyses: protocol for a modified e-Delphi study

PONE-D-22-18844R1

Dear Dr. Hunter,

We’re pleased to inform you that your manuscript has been judged scientifically suitable for publication and will be formally accepted for publication once it meets all outstanding technical requirements.

Kind regards,

Matthew Carroll, PhD., MEdL., MPod., BHSc

Academic Editor

PLOS ONE

Reviewers' comments:

Reviewer's Responses to Questions

**Comments to the Author**

1. Does the manuscript provide a valid rationale for the proposed study, with clearly identified and justified research questions?

Reviewer #1: Yes

2. Is the protocol technically sound and planned in a manner that will lead to a meaningful outcome and allow testing the stated hypotheses?

Reviewer #1: Yes

3. Is the methodology feasible and described in sufficient detail to allow the work to be replicable?

Reviewer #1: Yes

4. Have the authors described where all data underlying the findings will be made available when the study is complete?

Reviewer #1: Yes

5. Is the manuscript presented in an intelligible fashion and written in standard English?

Reviewer #1: Yes

6. Review Comments to the Author

You may also provide optional suggestions and comments to authors that they might find helpful in planning their study.

Reviewer #1: The authors have satisfactorily responded to the comments and revised the manuscript accordingly. Thanks.

7. PLOS authors have the option to publish the peer review history of their article (what does this mean?). If published, this will include your full peer review and any attached files.

Reviewer #1: **Yes: **Amédé Gogovor

---

## [Editor Report · Acceptance letter]

30 Sep 2022

PONE-D-22-18844R1 

Development of a checklist of standard items for processing individual participant data from randomised trials for meta-analyses: protocol for a modified e-Delphi study 

Dear Dr. Hunter:

I'm pleased to inform you that your manuscript has been deemed suitable for publication in PLOS ONE. Congratulations! Your manuscript is now with our production department. 

Kind regards, 

on behalf of

Associate Professor Matthew Carroll 

Academic Editor

PLOS ONE